# CeF$_3$-YF$_3$-TbF$_3$ Nanoparticle-Polymer–"Radachlorin" Conjugates for Combined Photodynamic Therapy: Synthesis, Characterization, and Biological Activity

Alexey Nizamutdinov [1,*], Elena Lukinova [2], Nail Shamsutdinov [1], Pavel Zelenikhin [1], Alina Khusainova [1], Marat Gafurov [1,*], Sergey Zinchenko [1], Damir Safin [1] and Maksim Pudovkin [1,*]

1  Institute of Physics, Kazan Federal University, 18 Kremlyovskaya Str., 420008 Kazan, Russia
2  Belgorod State National Research University, 85 Pobedy Str., 308015 Belgorod, Russia
*  Correspondence: anizamutdinov@mail.ru (A.N.); mgafurov@gmail.com (M.G.); jaz7778@list.ru (M.P.)

**Abstract:** Promising material for hybrid photodynamic therapy consisting of Ce$_{0.5}$Y$_{0.35}$Tb$_{0.15}$F$_3$ crystalline nanoparticles and Radachlorin is reported. One possible option of conjugation of Ce$_{0.5}$Y$_{0.35}$Tb$_{0.15}$F$_3$ nanoparticles and Radachlorin using polyethylenimine (PEI) is tested. The energy transfer reaches 28%. It is shown that conjugates of CeF$_3$—Tb$^{3+}$ NPs and Radachlorin using PEI—are stable, and the distance between nanoparticles and photosensitizer molecules is about 5 nm.

**Keywords:** photodynamic therapy; Radachlorin; nanoparticle-photosensitizer conjugates; flow cytometry

## 1. Introduction

Photodynamic therapy (PDT) is a modern and non-invasive form of therapy, used in the treatment of cancers and non-oncological diseases [1]. It is based on the use of photosensitizers that accumulate in pathological tissues and generate reactive oxygen species (ROS) when irradiated by the light of the appropriate wavelength [2]. The ROS are highly aggressive towards organic molecules, and the selectivity of their action is due to their low lifetime period. The phototoxic reactions occur only in the area of photosensitizer distribution, enabling selective destruction [3].

Nowadays, many types of photosensitizers are under investigation, and some already work within certified therapy protocols [4]. One of the critical characteristics of the photosensitizer is the wavelength of light which activates ROS generation because of practical demand of deeper propagation of light into the biological tissue. Photosensitizers being used now are activated in the red part of the visible spectral range or in the near-IR spectral range. Therefore, the conventional PDT can be used only for superficial diseases or mucous diseases where the light source can be delivered as a probe. There is an approach allowing use of X-ray or ionizing irradiation in order to overcome these limitations by creating conjugates of scintillator nanoparticles and photosensitizers [5,6]. The photosensitizers are loaded on the surface of the nanoparticle. During the irradiation by X-ray light, the nanoparticles convert the energy into visible light, which activates photosensitizer molecules located on the surface. The efficiency of energy transfer from scintillator nanoparticle to photosensitizers is a crucial factor for such an approach [3,6,7].

The Ce$_{0.5}$Y$_{0.35}$Tb$_{0.15}$F$_3$ fluoride material is well studied for different photonic applications [8,9]. In our previous work, the spectral–kinetic properties as well as morphology and chemical composition of these NPs was thoroughly investigated [8]. It was established that upon ultraviolet excitation, the NPs produce intense emissions in the visible range with lifetimes around 10 ms, which is the result of efficient nonradiative resonant energy transfer from Ce$^{3+}$ ions to Tb$^{3+}$ ions. It can be seen that Tb$^{3+}$-doped materials can potentially sensitize the chlorine-based PDT agents due to good overlapping of the absorption spectrum of the latter and emission of Tb$^{3+}$ ions. It was also discovered that the most

promising composition for use with chlorine e6 is $Ce_{0.5}Y_{0.35}Tb_{0.15}F_3$. In our work, we have investigated the chlorine e6-based commercially available photosensitizer Radachlorin [10], which is extensively investigated at the moment in various biomedical applications and utilized in clinical trials [11–13].

Thus, the main objectives of this study were to create conjugates of $Ce_{0.5}Y_{0.35}Tb_{0.15}F_3$ nanoparticles and molecules of photosensitizer Radachlorin, study such important parameters as the efficiency of energy transfer between doping ions and Radachlorin, and to study biological activity (cytotoxicity and cellular uptake).

## 2. Materials and Methods

### 2.1. Sample Preparation

All the chemicals used are of analytical grade. Citric acid ($C_6H_8O_7 \cdot H_2O$), $Y(NO_3)_3 \cdot 6H_2O$, $Ce(NO_3)_3 \cdot 6H_2O$, $Tb(NO_3)_3 \cdot 6H_2O$, $NH_4F$, ammonium, and polyethylenimine (PEI) were purchased from Sigma-Aldrich (St. Louis, MO, USA). All the chemicals were used without further purification. The doping ions' concentration is represented in molar percentage (mol.%). Crystalline $CeF_3$-$YF_3$-$TbF_3$ nanoparticles were fabricated via the co-precipitation method described earlier [14–16]. In more detail, $Ce_{0.5}Y_{0.35}Tb_{0.15}F_3$ nanoparticles were synthesized via the co-precipitation method in ammonium citrate solution with a 3-fold excess of $NH_4F$ fluorinating agent. Obtained nanoparticles were subjected to microwave treatment. In particular, a 0.3 M solution of citric acid was prepared. Then, the pH of the solution was adjusted to 5 by adding a 25% solution of ammonium in order to obtain an ammonium citrate solution (the pH value was controlled using the AMTAS PH-920 pH meter (USA). Then, rare-earth nitrates were added to the 150 mL ammonium citrate solution while stirring on a magnetic stirrer (400 rpm). The $NH_4F$ solution was added dropwise to the resulting mixture of rare-earth nitrates while stirring on a magnetic stirrer (400 rpm). The solution continued to be stirred at room temperature for 15 min. In the next step, the solution was treated by microwave irradiation (2.45 GHz, 650 W) in the microwave oven. The precipitate was purified with distilled water by centrifugation (Janetski K24; 3000–5000 rpm, centrifugation time was 10 min) 8 times. In order to form PEI–nanoparticle composites, 100 mg of dried nanoparticles were suspended in 10 mL of distilled water via sonication. Additionally, 25 mL of distilled water was added to 100 mg of PEI. The mixture was placed in an ultrasonic bath (model ODALQ40, 600 W, volume 4 L) for 7 min in order to obtain a homogenous solution. The colloidal solution of the nanoparticles was added dropwise to the PEI solution and stirred for several hours. Washing by centrifugation was carried out to remove residual unreacted PEI (until pH factor was about 6). We have used commercial PEI from Sigma Aldrich with average $M_w$ of about 25,000. As for the photosensitizer, we have used commercially available drug Radachlorin produced by RadaPharma (Moscow, Russia) company. It consists of 3.5 mg/mL water solution of a mixture of sodium salts of chlorine e6, chlorine p6, and purpurin 5 [11]. We diluted it in distilled water 1:10 for our experiments.

### 2.2. Sample Classification

Bold $CeF_3$-$YF_3$-$TbF_3$ nanoparticles are named NP, nanoparticles coated with PEI are named PEI-NP, and PEI-NP conjugated with "Radachlorin" are named PEI-NP-RCH.

### 2.3. Characterization of the Samples

The phase composition of the material was characterized by an X-ray diffraction method with Bruker D8 X-ray diffractometer (Cu, Kα radiation λ = 0.154 nm) (Billerica, MA, USA). The absorption spectrum of Radachlorin solution was obtained using a Schimadzu UV3600 spectrophotometer (Kyoto, Japan). The photoinduced luminescence spectra of the nanoparticles were recorded with a portable spectrometer EPP2000, StellarNet (Tampa, FL, USA). The luminescence decay of $Ce^{3+}$ and $Tb^{3+}$ ions was investigated with an MDR-23 monochromator and an FEU-87 photomultiplier tube with the time constant of about 6 ns.

The luminescence was excited by emission of the 4th harmonic of Q-switched YAG: Nd laser (wavelength 266 nm, pulse duration 10 ns, pulse repetition rate 10 Hz).

### 2.4. Cells Preparation and Cytotoxicity Assessment of CeF₃-YF₃-TbF₃ Nanoparticles

The A 549 (human lung carcinoma) cells were purchased in the Russian collection of vertebrate cell cultures, Russian Academy of Sciences, St. Petersburg, Russia. Lung carcinoma (A 549) cells were cultured in Eagle (MEM) biological medium with Hank's salts supplemented with 10% fetal calf serum (HyClone, Cytiva, Parramatta, Australia), glutamine (2 mM) (ReagentPlus, ≥99% (HPLC), (Sigma-Aldrich, St. Louis, MO, USA)), penicillin and streptomycin (100 IU/mL) at 37 °C in 5% $CO_2$ humidified atmosphere. The cytotoxicity of the samples was analyzed via the colorimetric MTT assay. The test protocol for cytotoxicity evaluation was adopted from elsewhere [17]. The A 549 cells were seeded in 96-well plates (SPL Lifesciences, Pocheon, Republic of Korea) in a concentration $10^4$ cells/well and incubated overnight. Then, we replaced the medium in wells with the 100 μL of fresh water containing nanoparticles. The sample suspension in distilled water was added to the cultural medium in a ratio of 1/10 (*v/v*) for each concentration. Then, the obtained suspension was sonicated for 10 min in a sonication bath (model ODA-LQ40, 600 W, volume 4 L, Oda Sevis, Moscow, Russia) until the suspension appeared homogeneous to the naked eye. The cells were treated with the samples at 0.01, 0.05, 0.1, 0.25, 0.5, and 1.0 g/L. Exposure time was 24 h at 37 °C in humid air (98%) containing 5% $CO_2$. Three hours before the end of the exposure period, MTT (3-(4,5-dimethyl-2-thiazolyl)-2,5-diphenyltetrazolium bromide, Sigma-Aldrich, #M5655, St. Louis, MO, USA) solution in 0.1 M pH 7.2 phosphate-buffered saline (PBS) (5 mg/mL, 20 μL/well) was added to the cells. After the completion of the exposure period, the supernatant was removed, and 100 μL/well solution containing 10% SDS (Sigma-Aldrich, #L3771) in PBS was added. Absorbance at 570 nm of each well was measured using a microplate reader (Biorad, xMark, Hercules, CA, USA). Each experiment was repeated 2 times, with five replications. The incubation time (120 min) for internalization study was measured between the moment of adding the samples and the moment of fixing by glutaraldehyde or between the moment of adding the samples and the moment of performing flow cytometry experiments.

### 2.5. Flow Cytometry

In this work, the possibility of the nanoparticles' internalization of the samples by the A 549 cells was investigated via flow cytometry. The cell monolayer with 80% confluence was trypsinized, and the cell suspension was precipitated by centrifugation (1500 rpm, 5 min). The supernatant was removed, the cells were resuspended in a complete Eagle (MEM) medium at a concentration of $10^6$ cells/mL, and a suspension of samples in distilled water in a ratio of 1/10 (*v/v*) was added. The final concentration of samples was 0.5 g/L for all cases. A cytometric assessment of the internalization of nanoparticles by cells was performed in a flow cytometer analyzing changes in the intensity of side-scattering signal (SSC) of the cells. The value of the SSC is proportional to the cells' granularity. FACSCanto II cytofluorometer (BD Biosciences, Franklin Lakes, NJ, USA) was used in the work. The initial processing of the results was performed using the FACSDiva Software v9.0 (BD Biosciences, Franklin Lakes, NJ, USA) program.

### 2.6. Transmission Electron Microscopy

Preparation of A549 cells treated with samples for TEM was carried out as follows. The cells were separated from the incubation medium by centrifuging for 5 min at 1000 rpm. Then, the supernatant containing non-cell-bound nanoparticles was removed, pellet was resuspended in PBS, and then centrifuged again, repeating it three times. After, the cells were resuspended in 1 mL of PBS and transferred to 1.5 mL Eppendorf tubes. The cells were centrifuged at 4000 rpm in an Eppendorf 5412 microcentrifuge (Hamburg, Germany) for 5 min. Then, the supernatant was carefully removed, and 1.5 mL of 2.5% glutaraldehyde (Merck, Rahway, NJ, USA) in PBS was added and left at + 4 °C. Samples (system "cells + PEI-

NP" as well as "cell + PEI-NP-RCH") were prepared for TEM in the following way. The samples were fixed overnight in 2.5% glutaraldehyde prepared in PBS at 4 °C, washed three times with 0.1 M phosphate buffer, and post-fixed by incubation in 1% (*w/v*) osmium tetroxide in the same buffer (25 mg/mL) at 4 °C for 4 h. The samples were dehydrated by passage through a graded ethanol series (30, 40, 50, 60, 70, 80, 90 and then 96% ethanol) before being transferred to 100% acetone and propylene oxide. Then, the samples were immersed in Epon resin (Fluka, Buchs, Switzerland) that contained propylene oxide added in the proportions (*v/v*) 1:2, 1:1, and 2:1, with each step involving a 12 h incubation. The samples were then embedded in pure Epon resin. Ultrathin sections (ca. 100 nm) were prepared using a glass knife on a Leica UC7 (Wetzlar, Germany), mounted on 200 mesh copper grids, and stained with 2% aqueous uranyl acetate (*w/v*) for 20 min and Reynolds' lead citrate (Reynolds 1963) for 7 min. The sections were examined using a transmission electron microscope (HT 7700 Exalens, Hitachi, Tokyo, Japan) operated at an accelerating voltage of 100 kV.

## 3. Results

### 3.1. Optical Spectroscopy of the Samples

The potential for effective energy transfer from $Tb^{3+}$-doped nanoparticles to chlorine e6-based photosensitizer is shown again for our objects of investigation. In Figure 1, the luminescence spectrum of our $CeF_3$-$YF_3$-$TbF_3$ nanoparticles is shown together with the absorption spectrum for Radachlorin. The absorption spectrum consists of familiar bands of chlorine e6. The Q band in our spectrum has two maxima due to presence of both e6 associates (shorter wavelength band) and noninteracting-with-each-other molecules (longer wavelength band) [18]. The group of $^5D_4$-$^7F_5$ transitions of $Tb^{3+}$ ions overlaps with the absorption band of Radachlorin, which poses the prospective efficient nonradiative energy transfer for application as the conversion from high energy photons to ROS generation and for investigation of nanoparticles + Radachlorin composite materials.

According to the XRD data, all the NPs are hexagonal-structured nanocrystals that correspond to the structure of matrices of $CeF_3$. Sharp peaks and lack of peaks from impurities are observed, suggesting the high purity of these samples. The possibility of conjugation between NPs and Radachlorin by means of PEI was investigated through spectral–kinetic characteristics' observation of colloidal solutions of samples. We used the luminescence decay curve of the $^5D_4$-$^7F_5$ transition of $Tb^{3+}$ ions at 541 nm as a marker when in a stepwise manner small portions of Radachlorin were added to the samples until colloidal solution precipitated. After each step, the luminescence decay of the $^5D_4$-$^7F_5$ transition of $Tb^{3+}$ was registered. The obtained luminescence decay curves are presented in Figure 2a,b.

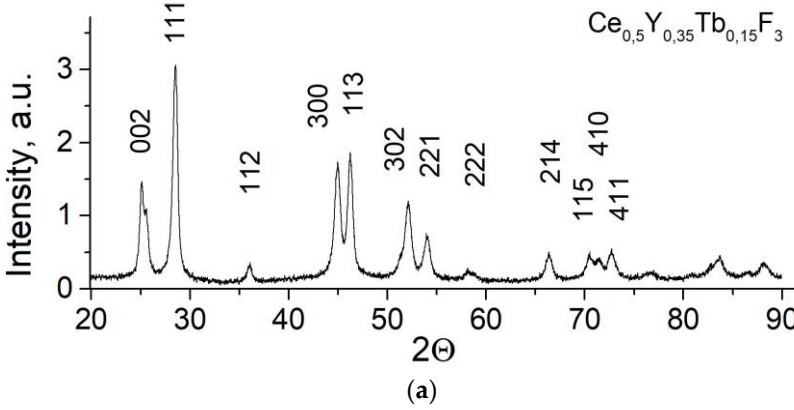

(a)

**Figure 1.** *Cont.*

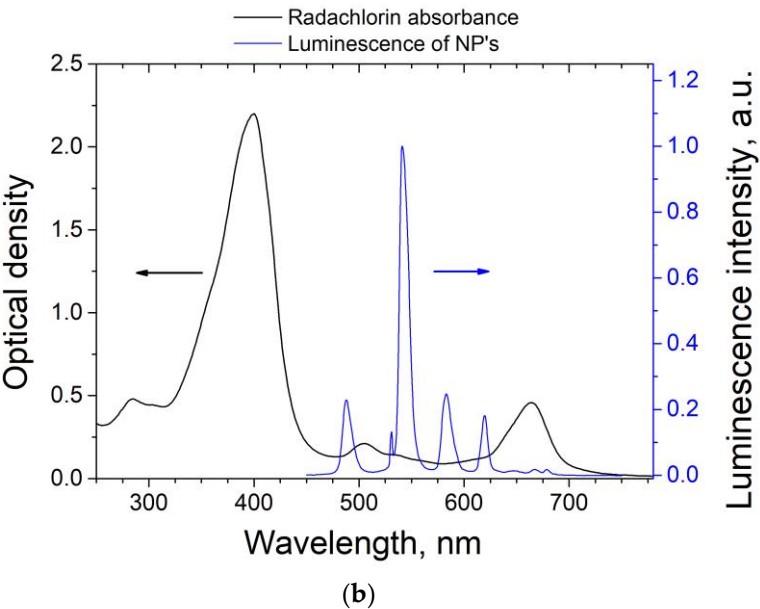

(**b**)

**Figure 1.** (**a**) XRD pattern of the studied sample, (**b**) luminescence spectrum of the colloid solution of $Ce_{0.5}Y_{0.35}Tb_{0.15}F_3$ nanoparticles covered with PEI under 266 nm excitation and absorption cross-section Radachlorin.

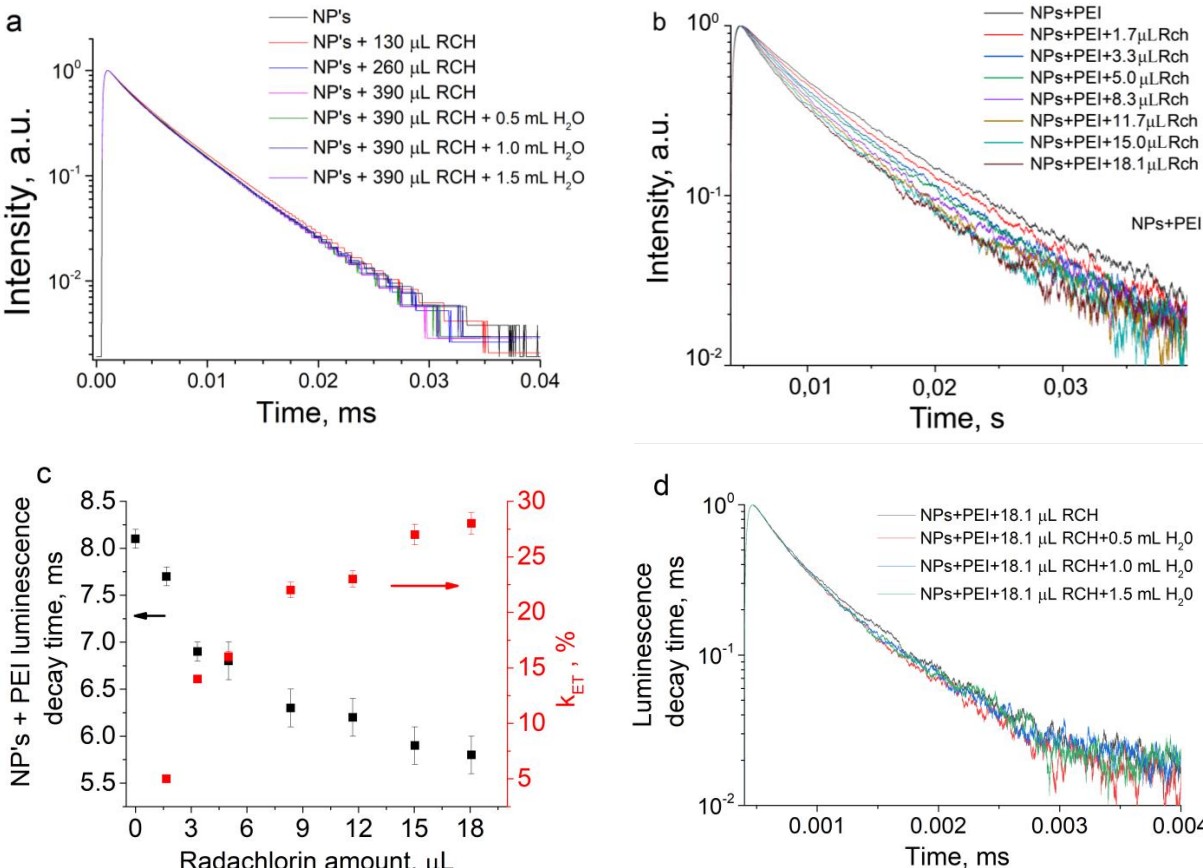

**Figure 2.** Luminescence decay of $^5D_4$-$^7F_5$ transition of $Tb^{3+}$ ions under 266 nm excitation in aqueous solution of $Ce_{0.5}Y_{0.35}Tb_{0.15}F_3$ nanoparticles bold (NP) (**a**) and covered with PEI (PEI-NP) (**b**) with Radachlorin photosensitizer added, (**c**) the luminescence decay of $^5D_4$-$^7F_5$ transition of $Tb^{3+}$ ions in PEI-NP with Radachlorin solution with water added, (**d**) dependencies of $Tb^{3+}$ luminescence decay time ($^5D_4$-$^7F_5$ transition) and energy transfer efficiency to Radachlorin.

Our results show that luminescence decay of NPs in colloid solution does not change with the addition of Radachlorin (Figure 2a), whereas we see the shortening of luminescence decay for PEI-NP in the presence of Radachlorin even for smaller amounts (Figure 2b). The data explicitly shows that the addition of Radachlorin leads to a decrease in decay time of $Tb^{3+}$ luminescence. This means that there is efficient energy transfer between NPs and Radachlorin. The behavior of the obtained decays appeared to be close to single-exponential for non-coated nanoparticles and significantly deviated from those coated with PEI, so the decay time was then estimated as an averaged decay time [19]:

$$t_{av} = \frac{\int t * I(t)dt}{\int I(t)dt} \qquad (1)$$

where $I(t)$ is the luminescence intensity. The results of lifetime evaluation are presented in Figure 2c. The value of luminescence of $Tb^{3+}$ ions at 541 nm in our colloidal solution without Radachlorin appears to be larger than that obtained by us for dry samples in our previous work [8], most probably due to radiative energy transfer between NPs. Then, the decay time values were used to calculate the efficiency of the energy transfer from NPs to Radachlorin. In order to estimate the efficiency of energy transfer ($k_{ET}$), a simple expression was used [20]:

$$k_{ET} = 1 - {\tau_{NPs}}\big/{\tau_{NPs + Rch}} \qquad (2)$$

where $\tau_{NPs}$ is the deacy time of $^5D_4$-$^7F_5$ transition of $Tb^{3+}$ ions in the absence of Radachlorin, and $\tau_{NPs + Rch}$ is the decay time of the same transmission but in presence of Radachlorin. The efficiency of energy transfer vastly increases at low amounts of Radachlorin, reaching the value of 20%. The highest observed value of energy transfer efficiency is 28% in the solution with 18.1 µL Radachlorin. The next step was to prove the formation of stable conjugates of NPs and Radachlorin molecules. The sample with NPs and 18.1 µL of Radachlorin was diluted with water three times in 0.5 mL increments, and decay time was monitored every time. The idea behind this approach is that the stable conjugates do not change the $Tb^{3+}$ decay time when diluted. On the other hand, if the NPs and Radachlorin molecules are not properly bonded, the additional water does increase average distance between NPs and Radachlorin, and energy transfer rate and the observed decay time increase. The results of the experiment are presented in Figure 2d. The amount of added water has little to no effect on decay time and energy transfer efficiency. This leads to the conclusion that stable conjugates of NPs and Radachlorin were formed by means of PEI. Our spectroscopy data allow estimation of the effective distance between $Tb^{3+}$ ions in NPs and Radachlorin molecules as elements of a conjugate. The Forster resonance energy transfer (FRET) theory is a reliable way to obtain this value [21,22]. FRET is a strongly distance-dependent transfer of energy between two elements usually called donor and acceptor. In order for FRET to happen, the donor should be an emissive molecule or particle and the acceptor should be able to absorb the light the donor emits. This energy transfer takes place at a distance range of approximately 1–20 nm. The key parameter in FRET theory is $R_0$, also called critical radius. This value demonstrates the distance between donor and acceptor, which provides 50% FRET efficiency. This parameter can be relatively easily calculated as [9]:

$$R_0{}^6 = \frac{2.07}{128\pi^5} \frac{\kappa^2 Q_D}{n^4} \int F_D(\lambda)\alpha(\lambda)\lambda^4 d\lambda \qquad (3)$$

where $\kappa^2$ is the orientation factor, $n$ is the refractive index of the medium, $Q_D$ is the luminescence quantum yield of the donor, $F_D$ is the donor emission spectrum normalized to unity ($\int F_D(\lambda)d\lambda = 1$), and $\alpha$ is the extinction coefficient of the acceptor. In the case under study for water solutions, the orientation factor can be averaged as 2/3; n is taken equal to the refractive index of water 1.333. The optical properties of the NPs have already been studied, and the results have been published [8]. Thus, the $Q_D$ value can be borrowed from the article as 83%. In the case under study, NPs serve as energy donors, whereas

Radachlorin is an acceptor. These data finally allow obtaining the $R_0 = 4.9$ nm for the pair NPs–Radachlorin. The next step is to estimate the distance between the NPs and Radachlorin in the obtained conjugates. It is well known that FRET efficiency can be calculated as [9]:

$$k_{FRET} = \frac{1}{1 + \left(\frac{r}{R_0}\right)^6} \qquad (4)$$

where $r$ is the distance between donor and acceptor. We consider that FRET is the only mechanism responsible for energy transfer between NPs and Radachlorin. Thus, Equations (2) and (3) effectively describe the same value. Taking that into account, it is now possible to calculate the distance between NPs and Radachlorin for different amounts of the added photosensitizer. Results of the calculation are presented in Table 1; the stable PEI-NP-Radachlorin conjugate corresponds to approximately 5 nm distance between $Tb^{3+}$ ions and chlorine e6 molecule.

**Table 1.** Distance between NPs and Radachlorin calculated from FRET.

| Sample | NPs+PEI 1.67 µL Rch | NPs+PEI 3.34 µL Rch | NPs+PEI 5.01 µL Rch | NPs+PEI 8.35 µL Rch | NPs+PEI 11.69 µL Rch | NPs+PEI 15.03 µL Rch | NPs+PEI 18.1 µL Rch |
|---|---|---|---|---|---|---|---|
| $r$, nm | 7.1 | 5.9 | 5.8 | 5.4 | 5.3 | 5.2 | 5.1 |

### 3.2. Cytotoxicity of PEI-NP and PEI-NP-RCH Samples

Relative viability histogram of A 549 cells (human lung carcinoma) treated with the studied samples in comparison to intact cells (control) is represented in Figure 3 (incubation time—24 h).

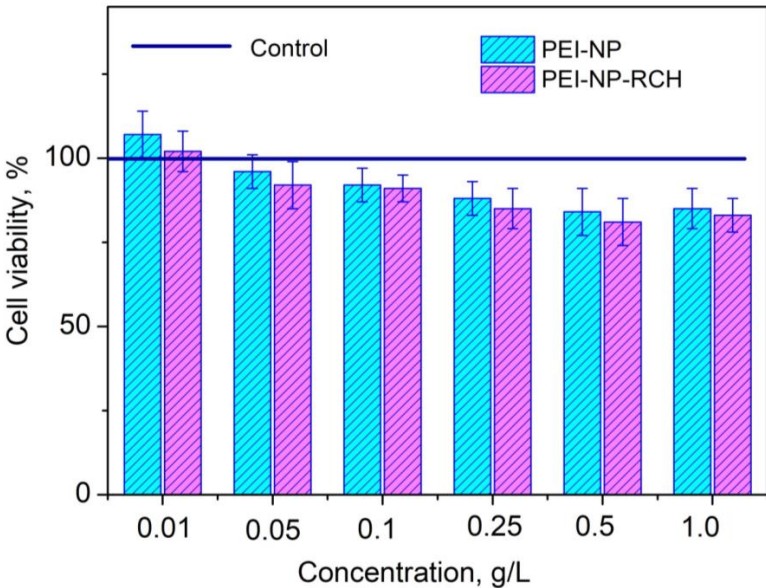

**Figure 3.** Relative viability of A 549 cells (human lung carcinoma) treated with the studied samples in comparison to intact cells (control cells) (incubation time—24 h).

It can be seen that all the samples demonstrate low cytotoxicity. The viability slightly decreases with the increase in concentration from ~ 100% (0.01 g/L) to ~90% (1.0 g/L). The samples do not demonstrate a difference in survival rate between each other. It can probably be explained by using biocompatible polymers as well as the clinically approved photosensitizer. The chosen concentration range is commonly used for different experiments where inorganic nanoparticles serve as drug carriers [23,24]. The obtained cytotoxicity data allow for the conclusion that the studied samples are promising for biomedical applications.

### 3.3. Visualization of Nanoparticles Internalization by A549 Cells Using TEM

TEM image of A 549 cells that are not exposed by the samples (control) is shown in Figure 4.

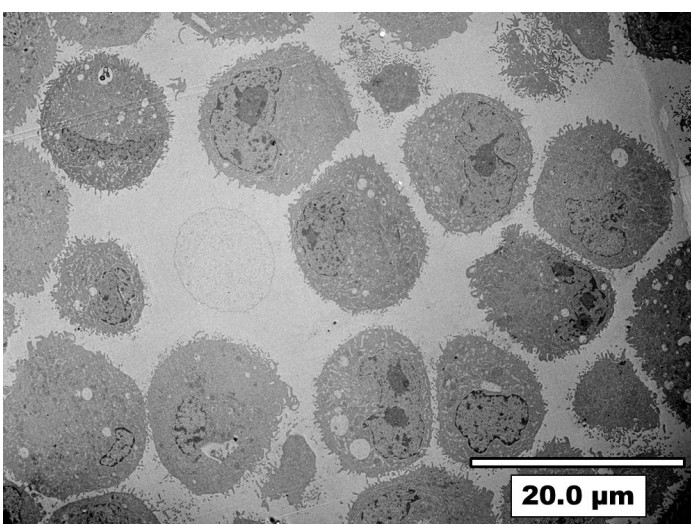

**Figure 4.** TEM image of A 549 cells (control).

It can be seen that the A 549 cells have a polygonal shape and sheet-like pattern, in normal monolayer culture, which is illustrative of epithelial cells including A 549 ones. The cell membrane has peculiar structural features such as membrane protrusions. Most of the membrane protrusions are planar folds (lamellipodia-like) with lengths from 200 to 800 nm. Plasma membrane extrusions have a diameter value in the 30–60 nm range. The same cell membrane structural features of A 549 cells were observed in works [25–27]. The cytoplasm contains vesicles having a diameter in the range from 200 to 2000 nm. Apparently, these vesicles are formed by the membrane protrusions that fused back into the membrane and trapped extracellular fluid (pinocytosis) [28]. The cell nuclei and other compartments are also clearly observed. It can be concluded that the morphology of A 549 cells is in agreement with literature analogs. Hence, the chosen cells can be used for further study of the cellular uptake process. TEM image of the A 549 human lung carcinoma cells treated by polymer-coated nanoparticle composite (PEI-NP) and TEM images of A 549 cells exposed by the PEI-NP-RCH (Figure 5a,b) for 2 h incubation reveal cell uptake of composites. More TEM images are represented in Supplementary Materials.

It can be seen that the PEI-NP composites and PEI-NP-RCH conjugates are effectively uptaken by A 549 cells. They are packed in the intricate shape vesicles floating in the cytoplasm. The linear size of these vesicles is in the 300–4000 nm range. The composites and the conjugates are not observed in the cell nucleus. It seems that coated fluoride nanoparticles are uptaken more efficiently in comparison to unmodified fluoride nanoparticles [26,27]. It can be suggested that the polymer coating and Radachlorin conjugation provide a positive zeta potential of nanoparticles [29]. It influences the interaction between positively charged nanoparticles and negatively charged cell membranes implementing the approaching of nanoparticles and cells. For the studied materials, it can be suggested that the internalization occurs via macropinocytosis classified as the endocytic process by which cells internalize fluids and particles together. During macropinocytosis, relatively large vesicles (0.3–5.0 µm linear size) are formed. The internalization seems to occur via planar folds, which are 100–300 nm in length. For the rest of the endocytic pathways (clathrin-mediated, caveolae-mediated, RhoA-mediated endocytosis) fewer vesicles (<200 nm) are formed; hence, these pathways were not considered. The linear sizes of vesicles are bigger than the linear sizes of planar folds. Probably, the coalescence of vesicles upon contact takes place. It seems that the vesicles are formed by these planar folds, which fused back into the membrane and trapped extracellular fluid containing cargo. Macropinocytosis is not directly coordinated by the presence of cargo unlike receptor-mediated endocytic

processes [30]. The membrane disruption was not observed for all the TEM images. It can be concluded that the studied materials do not destroy the cell membrane, which is crucially important for further therapeutic applications. Indeed, membrane disruption can cause necrosis, which is highly undesirable for therapy. On the other hand, drug carriers (including nanoparticle-photosensitizer conjugates) naturally uptaken by cells (by macropinocytosis or receptor-mediated pinocytosis) are able to give rise to apoptosis, which is a crucial factor for cancer therapy. It can be concluded that the composites and the conjugates are effectively uptaken by the cells after 2 h of incubation. Moreover, according to Figures 5 and S1, the samples were not found in the cell nuclei. It also confirms the safety of the samples. This fact paves the way toward therapeutic application of the studied materials.

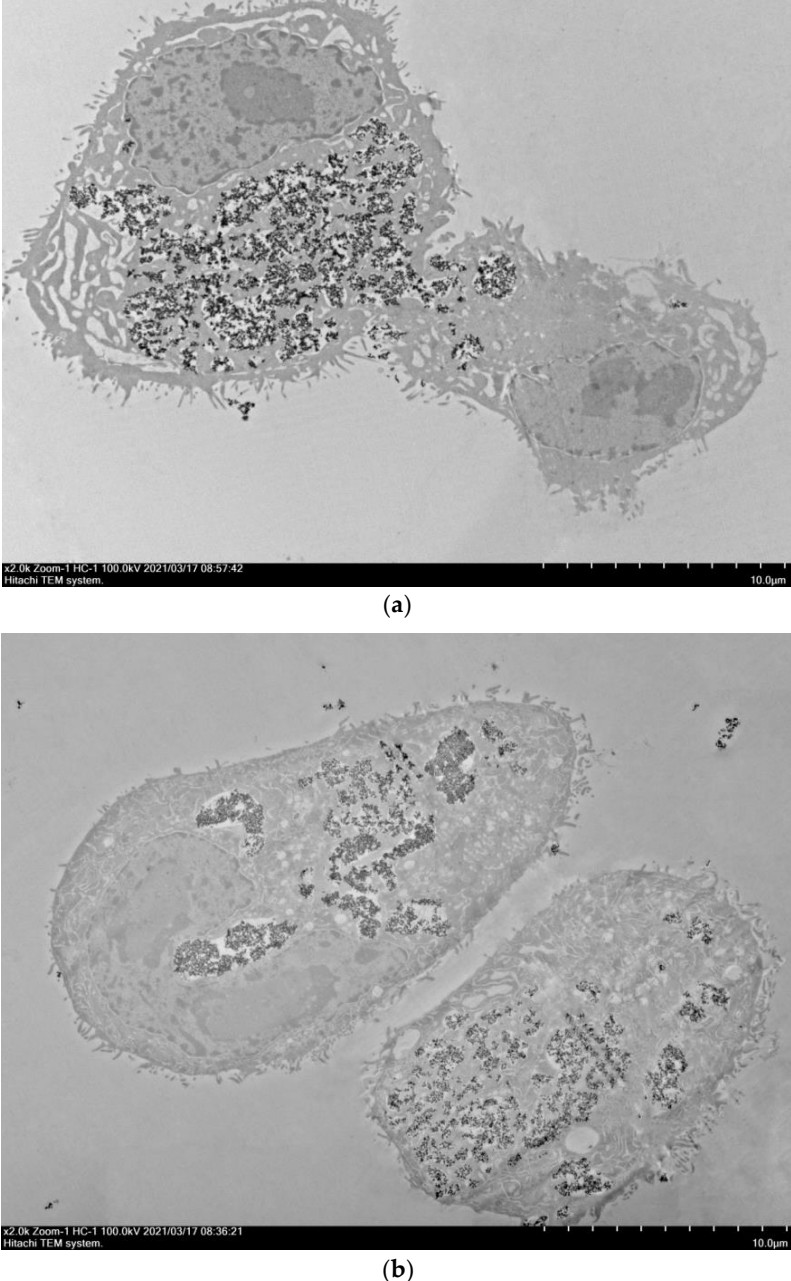

**Figure 5.** A 549 human lung carcinoma cells treated by polymer-coated nanoparticle composite (PEI-NP) (**a**) and TEM images of A 549 cells exposed by the (PEI-NP-RCH) (**b**) for 2 h incubation.

### 3.4. Flow Cytometry Study of Cellular Uptake of PEI-NP and PEI-NP-RCH

Although the TEM method allows visualizing the process of cell uptake, it is difficult to make a qualitative comparison of cell uptake efficiency for different types of nanomaterials. Such comparison requires analyzing a large number of TEM images to obtain appropriate statistics; however, the reliability of the data is still questionable. On the other side, the flow cytometry method allows studying up to $10^6$–$10^8$ cells in a sample, which is good for evidential statistics. In the flow cytometry method, one cell at a time goes through a laser beam, where the scattered light is a characteristic of the cells and their components [31]. Indeed, cells effectively scatters light due to a high cell optical inhomogeneity. The internalized particles (nanoparticles, composites, and conjugates) located in the cell and/or on the cell membrane increase the cell optical inhomogeneity that leads to an increase in side scattered light intensity (SSC) [32]. In this work, the SSC intensity serves as a measure of cell uptake efficiency. The SSC histogram for the studied composites and conjugates is represented in Figure 6. Note that for the control experiments, the cells were not exposed to the studied materials.

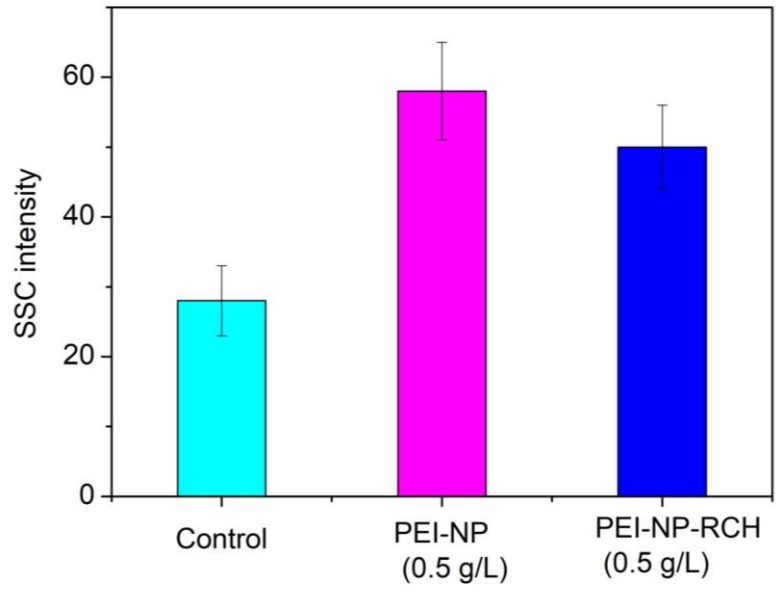

**Figure 6.** The intensity of SSC for different samples after 24 h of conjugates' exposure. The 0.5 g/L concentration was taken. The cells untreated with nanoparticles A549 used as control.

It can be seen that both PEI-NP and PEI-NP-RCH double SSC intensity in comparison to control. Indeed, according to [33], the PEI coating of $Er^{3+}$, $Yb^{3+}$:$NaYF_4$ fluoride nanoparticles provides 5 times higher positive zeta potential than PVP coating (51.1 and 10.2 mV, respectively). As it was mentioned above, the particles with high zeta potential effectively interact with the negatively charged cell membrane. Significant differences between the SSC intensity of cells treated with PEI-NP and PEI-NP-RCH were not found. We also estimated cytotoxicity of bare NPs, which was low. The results are represented in Table 2. The cytotoxicity of RCH was not estimated; however, according to the work [34], it is also low.

**Table 2.** Cytotoxicity of bare NPs.

| Concentration, g/L | Survival, % | Error Bar. % |
|---|---|---|
| 0 | 100 | 1.3 |
| 0.1 | 96.9 | 0.5 |
| 0.5 | 96.5 | 0.8 |

## 4. Conclusions

Effective energy transfer from NPs to Radachlorin molecules is demonstrated for NPs covered with PEI. As well as this, it is shown that energy transfer efficiency reaches 28% in the case of NP-PEI-RCH conjugates. Stability of NP-PEI-RCH conjugates is proved with distance between $Tb^{3+}$ ions and Radachlorin around 5 nm, calculated from spectroscopy data by Forster theory.

All the samples demonstrate low cytotoxicity (the viability slightly decreases with the increase in concentration from ~100% (0.01 g/L) to ~90% (1.0 g/L)). The samples do not demonstrate differences in viability between each other. It was suggested that low cytotoxicity is a consequence of the use of biocompatible polymers as well as the clinically approved photosensitizer. The chosen concentration range is commonly used for experiments.

It was shown that all the samples are easily uptaken by human lung carcinoma via macropinocytosis. The nanoparticles are packed in the intricately shaped vesicles floating in the cytoplasm. The linear sizes of these vesicles are in the 300–4000 nm range. The samples are not observed in the cell nucleus. The effectiveness of the uptake was estimated via flow cytometry. It was suggested that side-scatter signal intensity (SSC) is proportional to the effectiveness of the uptake.

Based on the facts, that the PEI-NP-RCH samples are non-toxic, easily uptaken by the cells, form stable conjugates, and demonstrate high energy transfer efficiency, it can be concluded that the studied PEI-NP-RCH samples are promising for biomedical applications including PDT.

**Supplementary Materials:** The following supporting information can be downloaded at: https://www.mdpi.com/article/10.3390/jcs7060255/s1. Figure S1: A 549 human lung carcinoma cells treated by polymer-coated nanoparticle composite PEI-NP and TEM images of A 549 cells exposed by the for 2 h incubation.

**Author Contributions:** A.N.—conceptualization, investigation, data curation, writing—editing; E.L. investigation and data curation; A.K.—investigation and data curation (optical spectroscopy); N.S.—investigation and data curation (biological activity); P.Z.—investigation and data curation (biological activity); M.G., investigation; S.Z. and D.S.—investigation and data curation; M.P.—investigation, data curation, writing—original draft, project administration. All authors have read and agreed to the published version of the manuscript.

**Funding:** This research was funded by the subsidy allocated to Kazan Federal University for the state assignment in the sphere of scientific activities (project number FZSM-2022-0021).

**Data Availability Statement:** Not applicable.

**Conflicts of Interest:** The authors declare no conflict of interest.

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
