# Peer review of "CeF3-YF3-TbF3 Nanoparticle-Polymer–“Radachlorin” Conjugates for Combined Photodynamic Therapy: Synthesis, Characterization, and Biological Activity"

_jcs, doi:10.3390/jcs7060255_

Round 1

Reviewer 1 Report

Hi, Authors.

The manuscipt have interest experimental data but need to carefully revision.

1. What is the content of cerium used in experiments? It`s more good to write the clear formulae.

2. Abstract has typo in 14 line, namely, CaF3: Tb. Is it CeF3?  

3. What is the molecular weight of PEI?

4. On page 2, line 83, may be typo. What is the solicitor used? It may be ultrasonic bath?

5. On page 2, line 89. What is the [Loschenov] reference?

6. I did not find the XRD results but diffractometer describing exist. Where is it?

7. The numbering of the pictures is off. It need renumbering.

8. On page 5, Fig a and b contain questions.

(a) The decay curve lower than x-axis

(b) Legend for y-axis contain typo, intencity, rather than intensity.

9. What is reason for choosing 18.07 microL of Rh?

10. Manuscript containt two version of writing (Rh and Rch). It need to be unified.

The manuscript need to reconsider after major revision.

Reviewer 2 Report

 The manuscript submitted by A. Nizamutdinov and co-workers describes the synthesis method and characterization for composite materials based on Tb-doped CeF3-YF3 nanoparticles conjugated with Radachlorin photosensitizer (PS). The authors declare the motivation of use Radachlorin as a low-toxic PS which is currently extensively investigated in different biomedical applications.

An authors declare that for efficient conjugation between NPs and PS, the synthesized core NPs should be covered with polyethylene (PEI). Authors employed an appropriate tool for characterization of the synthesized systems considering both optical properties and some biological diagnostics such as phototoxicity and cellular uptake assessed via tricky TEM imaging of the cells. Photoinduced luminescence utilized to estimate the efficiency of energy transfer between NPs and PS, as well as for estimation of the distances between NPs and PS through the FRET analysis.

In overall, I would highly evaluate the quality of the research reported in the manuscript and recommend this contribution for publication in J. Compos. Sci.

At the same time, I would encourage the authors address the following issue in order to improve the quality of the manuscript under revision or their further works on this topic.

1      1.    Few minor technical mistakes or typos should be addressed.

e.g. in line #5 in Abstract authors report the distance between NPs and PS as 5 A (angstroms), however from the text below (for example from table 1) it is clear that it should be 5 nm (nanometers).

Few language issues should be addressed:

e.g. in line #236 – “Taking that into the account” – no need to use “the”

e.g. in line #70  “0,3 M solution …” reported with comma; should be substituted by “0.3 M” etc

please make an additional proofread

2.    Reference list and quotes:

 In the introduction part some quotes have to be justified. For example, in line #28 authors quote single review dealing both with radiotherapy and photodynamic therapy (PDT). However, it will be more suitable to quote a couple of recent review which focused only on PDT.

 Similarly in a couple of sentence below (line #35) authors refer to X-ray photodynamic therapy (XPDT) and quote a well-cited but single experimental work. Following the same logic it would be more appropriate to cite here a couple of recent reviews in the field or in case of citing particular experimental work it seems to be more correct to mention along with recent work at least of the pioneer’s in the field.

 Finally, in the reference list up to 7 works of the first author of the present manuscript are mentioned. Please consider reducing this number by 4-5. At the same time, for example highly cited experimental work from Clement and co-workers (> 130 citations)  dealing with XPDT over CeF3-based conjugate is not mentioned (DOI: 10.1038/srep19954)

 3.    Remarks on Figure 2.

 The data reported in Fig. 2 should be put in order. The axis titles should be prepared in the same style (i.e. for panel (a), and (b,c) authors used different fonts.

Comma symbols should be removed from axis and legend of panel (a).

The reported data (a) should not go beyond the scale box.

Moreover, in line #92-93 authors provide abbreviation for composite as PEI-NP-RCH, while in panel (b,c) use Rh abbreviation.

It is also not clear why for experiments with NP’s panel (a) and PEI-coated NPs (b,c) the different amounts of PS were added into the system (i.e. hundreds and tens of RHC muL were added, respectively)? It will be easier if authors clarify this choice in the text or figure caption.

Line #205, since the authors declare that the higher energy transfer (estimated as 28%) was obtained for highest concentration of Radachlorin (18.07 uL), why they do not considered a further increase of the PS content in the system?

4.    Comment to Table #4 contest

The distance between NPs and PS molecules were estimated via FRET mechanism and the obtained values are in the range from 5-7 nm for different PS concentrations.

Also, by adding the water into the system as demonstrated in Fig 3c authors declare that PS molecules and NPs-PEI connected tightly to each other.

As far as I understand, then the 5-7 nm obtained from FRET analysis should assume that this spacing correspond to the thickness of PEI level on of the NPs surface.

If it is the case, the shell of thickness of 5-7 nm should be well-observed by means of HR-TEM or maybe even standard TEM imaging. i.e. the direct estimation of PEI thickness e.g. from TEM images might validate and reinforce the estimation of the NPs-PS spacing and provide a better understanding of the composite system for readers.

Anyway, some short discussion on the issue how authors interpret the obtained NP-PS spacing is highly appreciated

5.    Remarks on Figure 3. 

First, in the given version of the manuscript this image is mentioned as Fig. 1 most likely by mistake.

-        More typically concentration for this kind of plots usually reported in mug/ml

-        To see clearly the effect of PEI coverage as a biocompatible agent, the data for non-treated NPs (i.e. not covered with PEI nanoparticles) should be also reported

-        Also, it would be useful to report cytotoxicity of non-conjugated photosensitizer (Radachlorin) itself. Or please quote the reference (in case of any) where this issue has been studied and reported.

6.    Remarks on Figure 5, TEM images for cell with NPs (there is definitely smth wrong with numbering of the figures..)

1. The reported data looks very interesting and might be very useful for community working on the similar systems. Below in the text authors discussed that from statistical point of view SCC derived from flow cytometry provide more reliable data on cellular uptake.  

However, it would be of great interest if authors will provide more TEM photos (maybe in SI), i.e. 3-4 photos for PEI-NP and PEI-NP-RCH (in case of any) to demonstrate how the situation differs from cell to cell.

         2. In the discussion of obtained TEM images authors declare that coated NPs uptaken by the cells more readily (line #277-281). Authors explain + this by the modification of zeta potential for PEI coated NPs. In this case it might be of crucial importance if authors will measure zeta potential for bare and PEI-coated NPs and prove their hypothesis. It might dramatically improve the impact of the article! 

7.    Flow cytometry results

In the context of reported SSC author declare that NP-PEI and NP-PEI-RCH samples demonstrate roughly similar cellular uptake. Similarly, to the case of cytotoxicity assessment it would be curious to see the SSC value obtained for bare NPs. In the absence of zeta potential data, the lower SSC value obtained for bare NPs might somehow reinforce the interpretation of higher cellular uptake obtained for coated NPs.

Moreover, by varying concertation of NPs in the cell media (the date reported for 0.5 g/L) one might correlate how the concentration of NPs affects (or not affects) the cellular uptake.

Overall comments: since the described in the article objects is supposed to be considered as PDT agent, it would be nice to show the samples “in action”. Since the cytotoxicity of NPs+PS conjugate is low even at high concentrations (up to 1 g/L), it would be interesting to see the modification of the cell viability after exposure of the system with the same YAG:Nd laser that authors used for PL measurements. This will also significantly strengthen the impact of the work and might potentially bring more citations.

Reviewer 3 Report

This paper examines the conjugation of CeF3-YF3:Tb3+ nanoparticles and Radachlorin using polyethyleneimine (PEI). The paper contains observations but is not a complete study. I recommend that this paper not be accepted in the present form.

1) First of all, the paper contains weak, unclear research that is not sufficient to confirm the hypothesis or answer the question posed in the "Introduction" regarding the generation of reactive oxygen species (ROS) for photodynamic therapy (PDT).

2) I do not understand why the authors chose CeF3-YF3-TbF3 when there are many other nanoparticles with a better overlap of their emission with the absorption of Radachlorin. 

3) The luminescence is not studied in sufficient detail. There are no spectra of PEI-NP and PEI-NP-RCH samples. Photoluminescence excitation spectra are not studied. Why not? Excitation spectra can give very important information about the energy transfer process.

4) The decay time of CeF3-Y0.85Tb0.15F3 nanoparticles is more than 8 ms in this paper and 5 ms in [6] published by the same authors. This is not explained.

5) Why the authors concentrate on the Tb3+ luminescence and completely ignore the Ce3+ emission. In my opinion, the luminescence of Ce3+ ions in the UV-blue spectral region overlaps better with the absorption of Radachlorin. Therefore, the YF3 with a small amount of CeF3 should also be prepared and studied.

6) In the introduction, it was stated that when exposed to X-rays, the nanoparticles convert the energy into visible light, which activates photosensitiser molecules. Please try to do such research

7) The part about Forster resonance energy transfer (FRET) theory is completely wrong. Why do the authors calculate the distance between NPs and Radachlorin if this distance should be obtained experimentally using microscopy techniques? The calculation of the critical radius R0 is also incorrect. I cannot find the luminescence quantum yield in [6]. So, the authors should carefully study the literature on FRET.

8) The study is not finished. The scattered light intensity (SSC) did not give enough information. Also, PEI-NP and PEI-NP-RCH samples show similar behaviour. I propose to use special biomedical methods to study the effect of PEI-NP-RCH on A 549 (human lung carcinoma) cells. The authors showed that their PEI-NP and PEI-NP-RCH samples adhered (sticked) to A 549. This is not enough information. Maybe a lot of things also stick to A549. Maybe the tested nanoparticles do nothing for the generation of reactive oxygen species (ROS) for photodynamic therapy (PDT). So, the research is not complete.

9) There are many technical errors in the article. The numbering of the figures is strange and completely wrong. Figure 2 (a) is not shown correctly. There are problems with subscripts and superscripts in references. The English should be polished throughout the manuscript.

10) I am also interested in the authors' opinion on the current situation and illegal actions of their country. Is PTN huilo and ПТН ПНХ?

The article cannot be published in the present form because it contains many inaccuracies and research should be done in a more detailed and complete form.

Round 2

Reviewer 1 Report

Hi, authors.

The revised version of manuscript is more good but need to minor correct. 

1. The chemical formula (Ce0.5Y0.85Tb0.15F3) have mistake because substance is not electroneutrality. It need to correct.

2. In Table 1, the Radachlorin shortened to RC. Where is correct writing?

It ned to revised manuscript and correct.

Reviewer 2 Report

I am satisfied with the authors reply to the questions raised upon the revision procedure as well as modifications in the final version of the manuscript. The revised version of the manuscript is recommended for publication in J. Compos. Sci.

Reviewer 3 Report

The authors have submitted a revised version of the paper on the study of CeF3-YF3:Tb3+ nanoparticles and Radachlorin. Almost all my recommendations and comments have been ignored. The authors have not carried out the necessary experimental measurements. The study is not complete. I recommend that this manuscript should be rejected.

1) I do not understand why the authors focus on the Tb3+ luminescence and completely ignore the Ce3+ emission. Furthermore, in response to the second comment, the authors write that “CeF3 compound appears to be efficient scintillator for various types of ionizing radiation”. In my opinion, the Ce3+ emission at 330 nm should be better absorbed by Radachlorin because the optical density of Radachlorin in this region is close to 0.5, while the Tb3+ luminescence is located in the green spectral region where the optical density of Radachlorine is close to 0.1-0.2 (see Fig. 1 b). Therefore, three samples should be prepared and investigated: 1) YF3 with CeF3; 2) YF3 with TbF3; 3) CeF3- YF3- TbF3, which is under investigation. The results obtained for three samples should be compared and analyzed.

2) Fig. 1 shows the luminescence spectrum of the colloidal solution of YF3-CeF3:Tb nanoparticles covered with PEI. I asked for additional luminescence measurements. It was ignored. In my opinion, photoluminescence excitation spectrum can be registered for YF3-CeF3:Tb nanoparticles covered with PEI to check the energy transfer from Ce3+, host, maybe PEI to Tb3+. After that, in the next figure, the luminescence emission and excitation spectra of YF3-CeF3:Tb with PEI and Radachlorine should be given. Possible differences to the previous figure should be explained. In response to my comment, the authors write that the lifetime of Radachlorine is not long and the concentrations are very low. Lifetime does not usually affect the ability to record spectra. If Radachlorine has a weak influence on the Tb3+ spectra due to low content, it is worth to show this with relevant spectral data and an additional figure.

3) I proposed to do research showing that the studied nanoparticles convert X-rays into visible light, which activates photosensitizer molecules. This possibility is mentioned in the introduction of the manuscript. Unfortunately, the authors write in their reply about plans for future work. Thus, the hypothesis is not tested.

4) Regarding the application of the Forster resonance energy transfer (FRET) theory. I had written about the lack of luminescence quantum yield in [6]. The authors agree and reply that the quantum yield of Tb3+ can be estimated from luminescence decay data due to concentration quenching to be about 83%. What Tb concentrations were compared? Evidence and experiments were not provided in the revised paper.

I thought for a long time where 83% (high value for spin and parity forbidden Tb3+ transitions) came from and found that maybe 5 ms / 6 ms = 83% from [8] before [6]. But, it is impossible to compare. Furthermore, in response to the fourth comment, the authors explaining difference in luminescence lifetime agree: “In our work [6] we investigated dry samples. While here we investigate colloidal solution in water”. In my opinion, 0.5% Tb3+ in [8] before [6] is relatively high. Better will be the order of magnitude less. So, a colloidal solution similar to the one investigated, but containing a small amount of Tb like CeF3-Y0.9999Tb0.0001F3, should be prepared to compare lifetimes and extract the internal quantum efficiency of Tb3+. I write efficiency, not yield, because external quantum yield will be lower because some photons do not leave the sample. In this case, I recomend measure experimentally the quantum yield of CeF3-Y0.85Tb0.15F3 by absolute method using modern equipment with an integrating sphere.

Here, a new sample was meant only to evaluate the internal quantum efficiency, but new samples will still have to be prepared and all research must done with them from start to finish (see the first comment).

5) In the original manuscript, the authors showed that their nanoparticles adhered (sticked) to A 549 cells. I asked that maybe a lot of things also stick to A549. Maybe the tested nanoparticles do not contribute for the generation of reactive oxygen species (ROS) for photodynamic therapy (PDT). The authors reply that “the allogenic things were not adhered (sticked) to A 549 according to TEM images.” In my opinion, additional research is needed. The generation of ROS for PDT should be also investigated.

6) The authors have not clearly answered my last comment. It would be sufficient in the reply file to write briefly: "Yes, we agree". Then it would be clear that the authors are hostages of totalitarian circumstances. But, the authors decided to play dumb, which worsened the impression about them.

7) Technical errors have been fixed in the revised manuscript. But some inaccuracies remain. In particular, the energy transfer is 38% in "Abstract" and 28% in "Results".

8) An additional fact that is not related to the manuscript. The authors are from the Kazan Federal University (Russia). This university supports the Russia's invasion of Ukraine. See the official communiqué of the Academic Council of Kazan Federal University given in the official website: https://media.kpfu.ru/news/zayavlenie-uchenogo-soveta-kazanskogo-federalnogo-universiteta

Besides, in the press there are news that the staff visits the occupied territories, the letters "Z" and "V" have appeared on the university buildings and that students and staff are modernizing equipment for the military. Thus, the Kazan Federal University supports and promotes aggression and violence. This is unacceptable in the civilized world.

Summary of the review:

1) Methodological Flaws: The article has significant methodological flaws, such as incomplete data analysis and inadequate design of some experiments. The methodology is not rigorous enough to draw valid conclusions.

2) Insufficient Data and Analysis: The article presents insufficient data and incomplete analysis, leading to weak and inconclusive results. The data analysis is inadequate, and the article does not provide sufficient evidence to support the conclusions and interpretations. More detailed and complete research should be conducted.

3) Weak Results and Conclusions: The results presented in the article are inconclusive and do not support the research question or hypothesis. The conclusions drawn from the results are not supported by the data.

4) Overall Weakness: The article overall lacks quality, rigor, and scientific merit. The research is not well-designed and well-executed, and the article does not meet the standards expected for publication in a reputable scientific journal.

Conclusion – reject.

Round 3

Reviewer 1 Report

Hi, Authors.

I was decide to accept the revised version of manuscript.

Reviewer.

Reviewer 3 Report

As the reviewer, I provided several comments and recommendations to improve the quality of the paper, but it seems that only 10 % of them have been addressed in the second revised version. The authors have not carried out the necessary experiments. The research is not complete. After careful consideration, I regret to inform that I cannot recommend this manuscript for publication. I recommend that this manuscript should be rejected.

In particular, I suggested to examine the role of Ce3+ ions besides Tb3+; to study better the luminescence and the quantum yield; to check the activation of photosensitiser molecules under X-rays; to test the generation of reactive oxygen species (ROS) for photodynamic therapy (PDT) and some other recommendations in the first and second review report. My comments are very important because they address key aspects of the paper, such as the methodology, data analysis, and interpretation of results. However, my recommendations have been ignored.

Thus, the article has incomplete data analysis and poor design of some experiments, leading to weak and inconclusive results. More detailed and complete research should be carried out. The research is not well conducted, and the article does not meet the standards required for publication.

Therefore, I recommend that the manuscript must be rejected.